# Association of thoracic spine deformity and cardiovascular disease in a mouse model for Marfan syndrome

**Rodrigo Barbosa de Souza**[1], **Luis Ernesto Farinha-Arcieri**[1], **Marcia Helena Braga Catroxo**[2], **Ana Maria Cristina Rebelo Pinto da Fonseca Martins**[2], **Roberto Carlos Tedesco**[3], **Luis Garcia Alonso**[3], **Ivan Hong Jun Koh**[4], **Lygia V. Pereira**[1] *

1 University of São Paulo, Department of Genetics and Evolutionary Biology, São Paulo, SP, Brazil, 2 Biologic Institute of São Paulo, Department of Electron Microscopy, São Paulo, SP, Brazil, 3 Federal University of São Paulo, Department of Morphological and Genetics, São Paulo, SP, Brazil, 4 Federal University of São Paulo, Department of Surgery, São Paulo, SP, Brazil

* lpereira@usp.br

## Abstract

### Aims

Cardiovascular manifestations are a major cause of mortality in Marfan syndrome (MFS). Animal models that mimic the syndrome and its clinical variability are instrumental for understanding the genesis and risk factors for cardiovascular disease in MFS. This study used morphological and ultrastructural analysis to the understanding of the development of cardiovascular phenotypes of the the mgΔ$^{loxPneo}$ model for MFS.

### Methods and results

We studied 6-month-old female mice of the 129/Sv background, 6 wild type (WT) and 24 heterozygous animals from the mgΔ$^{loxPneo}$ model. Descending thoracic aortic aneurysm and/or dissection (dTAAD) were identified in 75% of the MFS animals, defining two subgroups: MFS with (MFS$^+$) and without (MFS$^-$) dTAAD. Both subgroups showed increased fragmentation of elastic fibers, predominance of type I collagen surrounding the elastic fiber and fragmentation of interlaminar fibers when compared to WT. However, only MFS animals with spine tortuosity developed aortic aneurysm/dissection. The aorta of MFS$^+$ animals were more tortuous compared to those of MFS$^-$ and WT mice, possibly causing perturbations of the luminal blood flow. This was evidenced by the detection of diminished aorta-blood flow in MFS$^+$. Accordingly, only MFS$^+$ animals presented a process of concentric cardiac hypertrophy and a significantly decreased ratio of left and right ventricle lumen area.

### Conclusions

We show that mgΔ$^{loxPneo}$ model mimics the vascular disease observed in MFS patients. Furthermore, the study indicates role of thoracic spine deformity in the development of aorta diseases. We suggest that degradation of support structures of the aortic wall; deficiency in the sustenance of the thoracic vertebrae; and their compression over the adjacent aorta

**Data Availability Statement:** All relevant data are within the manuscript and Supporting Information files.

**Funding:** Souza, RB, Farinha-Arcieri, LE, Caroxo, MHB, Martins, AMCRPF, Tedesco, RC, Alonso, LG, Koh, IHJ and Pereira LV were financed in part by the Coordenação de Aperfeiçoamento de Pessoal de Nível Superior - Brazil (CAPES) – Finance Code 001; and by Fundação de Amparo à Pesquisa do Estado de São Paulo.

**Competing interests:** The authors have declared that no competing interests exist.

resulting in disturbed blood flow is a triad of factors involved in the genesis of dissection/aneurysm of thoracic aorta.

## 1. Introduction

Marfan syndrome (MFS) is an autosomal dominant disease of the connective tissue, with considerable clinical variation [1, 2, 3]. The disease is caused by mutations in the *FBN1* gene, which encodes fibrillin-1, the most abundant glycoprotein of the microfibrils of extracellular matrix (ECM) [4,5,6]. Fibrillin-1 microfibrils are the scaffold for elastin deposition, forming the elastic fibers, a major component of the aortic wall [4,5].Thus, fibrillin-1 dysmorphism is considered to be the primary causal factor for the cardiovascular disease's related morbidity and mortality in MFS.

Mouse models of MFS have been instrumental in elucidating the complex molecular mechanisms underlying the disease. In particular, the $mg\Delta^{loxPneo}$ model for MFS has an in-frame deletion of exons 19–24 leading to the production of an internally truncated fibrillin-1 protein [7]. Heterozygotes in the Sv/129 isogenic background present MFS-phenotypes including kyphosis and elastic fiber disruption and thickening of the aortic media, with high phenotypic variability [7]. The aim of this study was to further characterize cardiovascular phenotypes and their variability in this model. We identify other cardiovascular phenotypes and alterations in blood flux in the aorta in a subset of heterozygotes, and show a correlation between these phenotypes and thoracic spine tortuosity.

## 2. Materials and methods

### 2.1 Animals

Thirty four 6-month-old female mice of the 129/Sv background were used. Of these, 10 animals were Wild type (WT) and 24 were heterozygotes from the $mg\Delta^{loxPneo}$ model [7]. This study was approved by the Institutional Animal Care and Use Committee of the Instituto de Biociências at the Universidade de São Paulo. Protocol ID: CEA/IBUSP 272/2016 Process 16.1.632.41.7.

### 2.2 Morphological & morphometric study

Aorta analysis: Thoracic aorta samples from the third to the eight thoracic vertebrae ($T_{III}$-$T_{VIII}$ region) were collected, fixed in 2.5% glutaraldehyde in 0.1 M sodium cacodylate buffer (pH 7.4) and embedded in resin (Technovit Kit 7100). Four-micron-thick transversal slices were cut, stained with Toluidine Blue, Periodic acid-Shiff (PAS), Weigert and Picrosirius Red, and were examined with a Carl-Zeiss Axio Scope Microscope.A1. Images of four different points of each of five transversal sections of the aorta of each animal were taken at 40× magnification. The area of the lumen of the aorta was measured using the "contours" tool of the ZEN software. Thickness of the elastic fibers was measured and median per animal was calculated. In addition, intensity of red collagen fiber staining and adherence of the interlaminar fibers were analyzed. Three animals of each MFS subgroup were perfused after sacrifice with 4% paraformaldehyde (PFA) in 0.1 M sodium cacodylate buffer (pH 7.4) for 30 minutes. Subsequently, the aorta was isolated and samples were fixed by immersion in the same fixative for 48h. Before inclusion, the samples were descaling with the decalcification solution (PFA 4%, EDTA 5% in 0.1 M sodium cacodylate buffer (pH 7.4)) for 20 days, and embedded in resin (Technovit Kit

7100). Four-micron-thick transversal slices were cut, stained with Toluidine Blue, and were examined with a Carl-Zeiss Axio Scope Microscope.A1. Images were taken at 2.5× magnification.

Heart analysis: Dissected hearts were cut transversally in half, fixed in 4% paraphormalde-hyde in 0.1 M sodium cacodylate buffer (pH 7.4), and embedded in resin (Technovit Kit 7100). Four-micron-thick transversal slices were cut and stained with Toluidine Blue. Images were taken at 2.5× magnification with a Carl-Zeiss Axio Scope Microscope.A1. Mounted the pictures were used to measure total heart area, and wall-thickness and lumen-area of right and left ventricles.

## 2.3 Electron microscopy

Aortic samples were fixed in 2.5% glutaraldehyde in 0.1 M sodium cacodylate buffer (pH 7.4) and after with 1% osmium tetroxide in 0,1M sodium cacodylate buffer. Then, samples were immersed in 2% lead citrate in 0,1M sodium cacodylate buffer, resin embedded (eSpurr) and examined with a PHILIPS EM 208S® electron microscope.

## 2.4 Anatomical analysis of the thoracic spine

Before sacrifice for collection of tissue samples, mice were anesthetized with 0.01/100mg Keta-mine® and Xylazine® (4:1) by intraperitoneal route, and fixed in lateral decubitus with the aid of adhesive tape. Digital radiographic images were obtained with the In-vivo Imaging System FX PRO (Carestream Molecular Imaging). To assess the severity of the thoracic vertebra deformation, Kyphosis Index Ratio (KI) was used [7,8]. Briefly, KI is the ratio between the length of a straight line from the last cervical vertebra to the sixth lumbar vertebra and the length of a line perpendicular to the latter, from the dorsal edge of the vertebra at the point of greatest curvature to the first line.

## 2.5 Aortic blood flow

Under general anesthesia (0.01/100mg Ketamine® and Xylazine® (4:1)), the abdominal aorta was dissected above the infra-phrenic artery near aortic hiatus in order to separate it from the spine ligaments, and the ultrasound flowprobe 2SB/T206 (Transonic Systems Inc, Ithaca, NY) was placed around the vessel. The surgical procedure was performed under the stereo microscope M900 D.F. Vasconcellos®.

## 2.6 Statistical analysis

The results were analyzed by Kruskal-Wallis and Mann Whitney and Pearson's correlation, and point biserial correlation tests [9]. Differences were considered significant at $p < 0,05$. The minimal data set is available as Supporting Information.

# 3. Results

## 3.1 Aorta characterization

Histologic analysis of the thoracic aorta detected presence of descending thoracic aortic aneu-rysm and/or dissection (dTAAD) in 75% (18/24) of MFS and in none of the wild type (WT) animals. From those, 83.3% (15/18) presented aneurism and 33.3% (6/18) aortic dissection with false lumen and rupture of the tunica intima. (Fig 1).

In addition, we measured the area of the aortic lumen in WT and MFS animals with and without dTAAD (Fig 1B). We observed a significant decreased of aorta's lumen in MFS ani-mals with dTAAD when compared to WT and MFS animals without dTAAD. Based in these

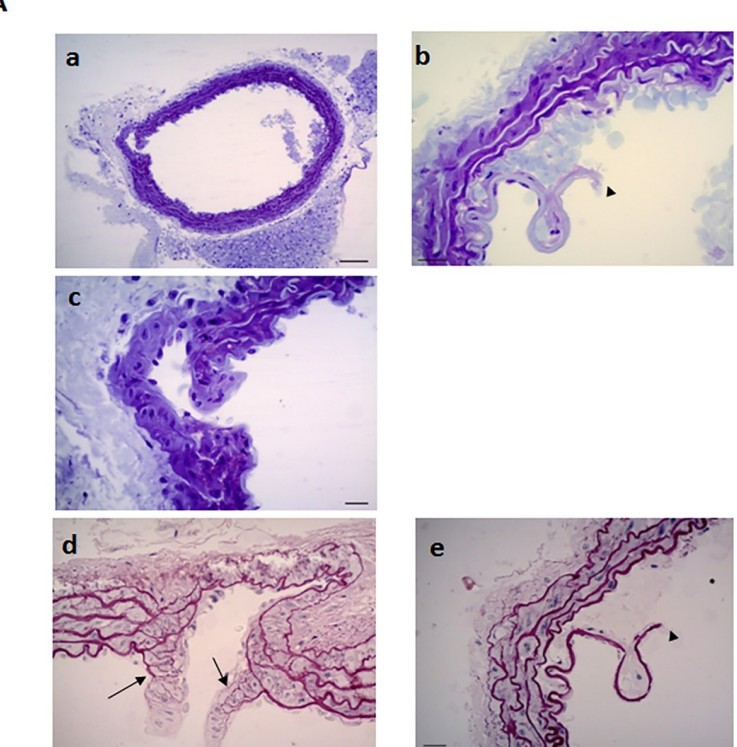

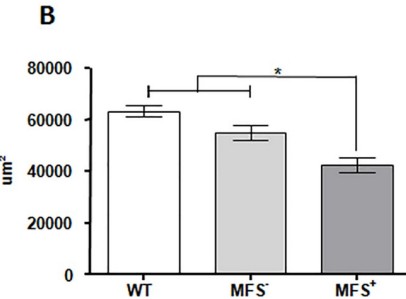

**Fig 1. Aorta characterization.** (A) Representative images of aortas from MFS⁺ animals with aneurysm (a, c and d) and aortic dissection (b and e). Aortic samples were analyzed by Blue Toluidin staining (a, b and c) and Weigert Staining (d and e). Aneurismal regions (d) showed thinning of the elastic fibers (→). Rupture of the intimal lamella (►) was observed in aortic dissection (d and e). Scale bar 50µm (a), Scale bar 10µm (b,c and d). (B) Area of aortic lumen in the three different groups of animals.(*) p<0,05.

findings, we focused in the comparison between MFS animals with (MFS⁺) and without (MFS⁻) dTAAD.

### 3.2 Elastic fibers

Histological analysis of the tunica media demonstrated multiple elastic fiber fragmentations and focal absence of elastic fibers in the media-lamella of MFS⁻ and MFS⁺ mice (Fig 2A, panels i,ii,iii). The TEM analysis revealed organized and continuous elastic fibers in WT animals, while those fibers were disorganized and with multiple ruptures in both of MFS subgroups (Fig 2A, panels iv, v,vi). Furthermore, the difference of the mean elastic fibers thickness

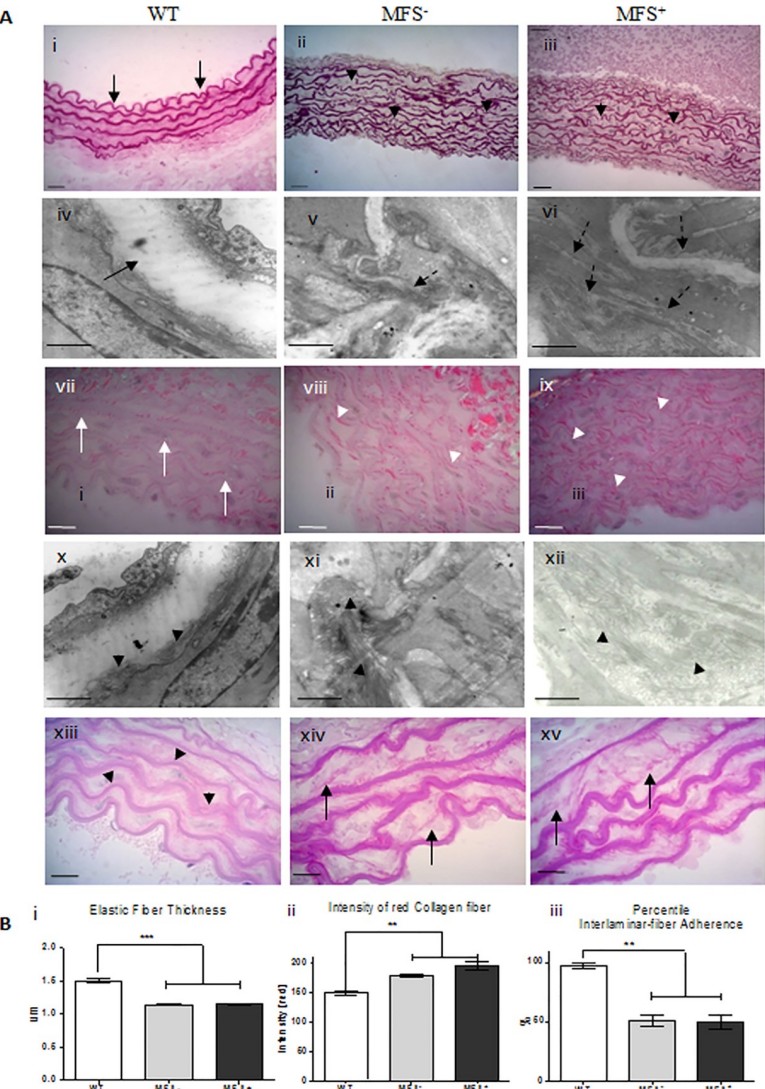

**Fig 2. Histological analysis of the aorta.** (A) (i, ii, iii) Weigert Staining of the tunica media showed elastic fiber fragmentation in MFS animals (black arrowheads); (iv, v, vi) TEM analysis showing micro-fragmentation of the elastic fiber in MFS animals (dashed arrow); Both techniques showed continuous elastic fibers in the WT samples (i, iv, arrow); (vii, viii, ix) Picrossirus red staining showed little prevalence of type I collagen surrounding the elastic fibers in WT samples (white arrows). In contrast, MFS- and MFS+ mice showed type I collagen clusters (red) (white arrowheads). Results were confirmed by TEM (x, xi, xii, arrowheads). In addition, the analysis of the interlamellar fibers (xiii, xiv, xv) revealed clustered fibers attached to the elastic fibers in WT mice (arrow heads) and fragmented and sparse fibers in MFS- and MFS+ mice (arrows). **(B)** (i) quantitative analysis of elastic fiber thickness in WT and MFS thoracic aorta showing a significant reduction of fibers thickness in MFS mice (*** $\rho<0.0001$).(ii) Picrossirus red analysis showing a significant increase of the intensity red fiber color in MFS- and MFS+ when compared to WT animals (** $\rho<0.005$). (iii) Quantification of interlaminar fibers revealing significant decreased in adherence of inter-laminar fibers in both MFS+ and MFS- when compare to WT animals (** $\rho<0.005$). Scale bars 10μm (A- i, ii, iii;, vii, viii, ix, xiii, xiv and xv); 200nm (A- iv, v, vi, x, xi and xii).

between, WT (1,50 μm ± 0.26,) and MFS (MFS-: 1.14μm ± 0.34; MFS+: 1.15 μm ± 0.34) was highly significant ($\rho<0,0001$), while no difference was observed between MFS+ and MFS- sub-groups. (Fig 2B, panel i). These findings reflect a highly compromised elasticity in the aortic wall of MFS mice regardless of the presence or absence of the vascular disease.

### 3.3 Collagen fibers

Histochemical analysis of collagen fibers showed a significantly higher predominance of collagen type-I surrounding the elastic fibers in MFS animals in comparison to WT (Fig 2A, panels vii, viii, ix; Fig 2B, panel ii). The TEM analysis showed scattered collagen clusters interspersed among the fragmented elastic fibers in both MFS subgroups (Fig 1A, panels xi, xii). In contrast, in WT animals collagen fibers were situated adjacent to the elastic fibers (Fig 2A, panel x). The high prevalence of the collagen type I in both MFS subgroups suggests a higher degree of stiffness in the aorta wall of MFS mice.

### 3.4 Inter-lamellar fibers

In the WT group the inter-lamellar fibers were adhered to the elastic fibers (Fig 2A, panel xiii), whereas in the MFS subgroups the fibers were mostly detached. (Fig 2A, panels xiii, xiv, xv; Fig 2B, panel iii). These findings indicate an unstable structure of the elastic fibers of the mid-layer of the aorta in MFS mice, impairing their designed physiological function.

### 3.5 Kyphosis index

Kyphosis Index (KI) was used to measure the degree of the thoracic spine deformity. We observed a significant KI reduction only in MFS$^+$ animals when compared to WT ($\rho < 0.0079$) and MFS$^-$ animals ($\rho < 0.0035$) (Fig 3A and 3B). In contrast, KIs did not differ significantly between MFS$^-$ and WT groups. These findings suggest a connection between dTAAD and thoracic vertebral deformity.

In order to test a correlation between the skeletal phenotype and dTAAD, we applied the point biserial correlation test that combines categorical (presence or absence vascular disease) and continuous (KI) variables [9]. We found a correlation of -0,91 (p<0,0001), which indicates that the more severe the defect of the spine, the greater the chance of vascular disease in the thoracic aorta (Fig 3B).

### 3.6 Association between thoracic spine deformity and thoracic aorta shape

Visual inspection of the anatomy of the thoracic aorta showed that the MFS$^+$ subgroup presented greater aorta tortuosity compared to MFS$^-$ and WT (Fig 3C, panels i, ii, iii). In addition, histological analysis of cross-sections of the thoracic aorta revealed that MFS$^+$ animals had dysmorphic aorta lumen which contrasted with the round shape of the vessel in WT and MFS$^-$ mice (Fig 3C, panels iv, v, vi). We performed the same analysis in 3 perfused MFS-animals of each group to evaluate the *in vivo* shape of the vessel, and confirmed the dysmorphism of the aorta only in MFS$^+$ mice (Fig 3C, panels vii, viii).

Finally, using Pearson analysis we found a positive correlation between KI and area of the aortic lumen (cor 0.7984, p<0.0001; Fig 3D), showing more severe spine tortuosity is associated with smaller aortic lumen area. These results suggested a possible interference of deformation of the thoracic spine on the aorta shape which may be expected since the thoracic aorta descends continuously and very close to the thoracic spine.

### 3.7 Aortic blood flow

The aortic deformations described above could lead to disturbances in blood flow. We used spectral curve analysis of the aorta to test this hypothesis. The analysis was performed in the abdominal aorta, close to the aorta hiatus, before the first ramification of the abdominal aorta (the infra-phrenic artery). Thus, alterations in blood flow in this region reflect those in the thoracic region.

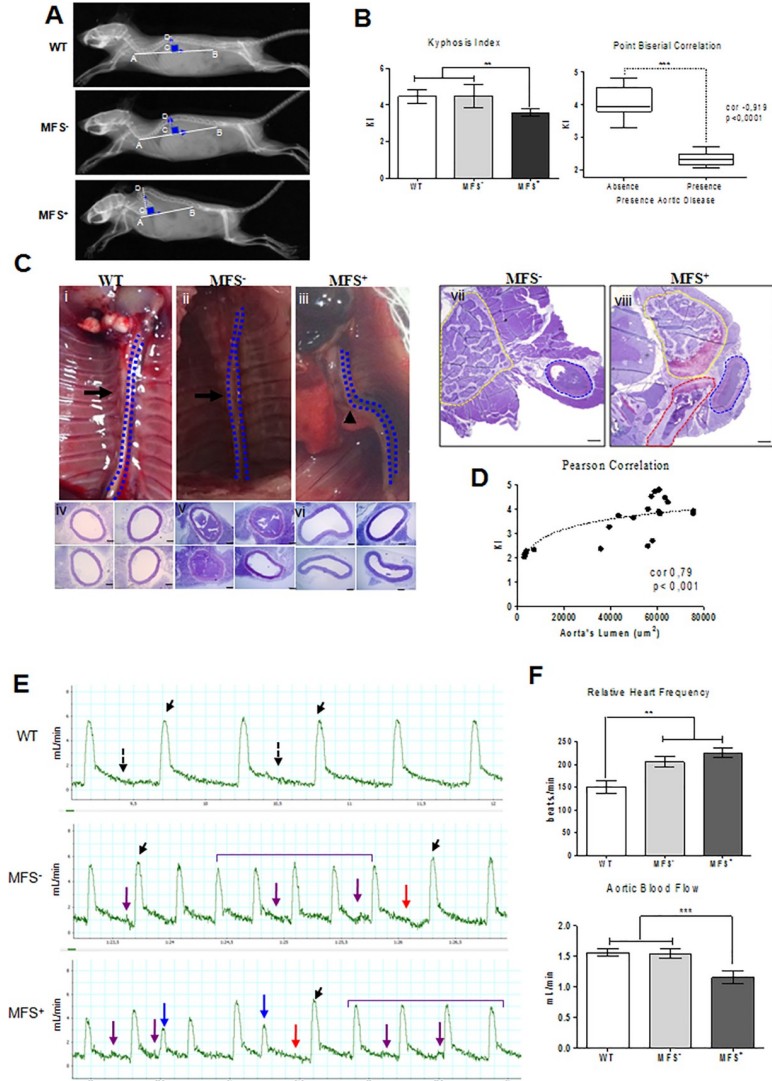

**Fig 3. Relationship between KI and vascular disease.** **A**. Representative X-ray images of WT and MFS⁻ or MFS⁺ mice showing KI measurement. **B.** Analysis of KI showing (left) a significant reduction in MFS⁺ when compared to WT and MFS⁻ (** ρ<0.005); and (right) correlation between presence of aortic disease and lower KI values (*** ρ<0.0001). **C.** (i-iii) Anatomical aspects of the thoracic aorta (contour in blue). Histological analysis showing that (i-vi) WT and MFS⁻ aortas have a circular appearance, while MFS⁺ presented dysmorphic aortas, and (vii-viii) dysmorphism of the aorta in perfused MFS animals. Dashed lines indicate aorta (blue), thoracic vertebra (yellow) and thoracic rib (red). **D.** D. Correlation between KI and area of the aortic lumen (cor 0.7984, p <0.0001). **E.** Spectral curve analysis of aortic blood flow. WT animals present uniform monophasic flow pattern with similar peak systolic velocities (PSVs) (black arrows) and no alterations in relaxation curves (dotted arrow). In MFS⁻ we observe a PSV (black arrow), arrhythmic spikes (purple bracket), compensator steady state (red arrow) and broadened waveform in diastole (purple arrow). In MFS⁺ animals, the monophasic flow showed varying PSVs, arrhythmic spikes (purple bracket), extra-systoles (blue arrow), compensatory stead spikes (red arrow) and broadened waveform in diastole (purple arrow). **F.** Quantification of heart beat rates revealed significantly increased heart rates in both MFS subgroups when compared to WT animals (** ρ<0.005). Quantification of aortic blood flow showed a significant reduction in MFS⁺ when compared to WT and MFS⁻ (*** ρ<0.0001). Scale bars 200μm (C- iv, v and vi).

In MFS⁺ animals, the monophasic flow showed varying peak systolic velocities (PSVs), arrhythmic spikes, extra-systoles, compensatory stead spikes and broadened wave in diastole (Fig 3E). In contrast, the WT group showed a uniform monophasic flow pattern with similar PSVs and no alterations in relaxation curve patterns (Fig 3E). Animals in the MFS⁻, group also

presented a uniform monophasic flow pattern with uniform PSVs, but exhibited arrhythmic spikes, compensatory stead state and broadened curve wave in diastole (Fig 3E, panel ii).

Finally, we observed significantly increased heart rates in both MFS subgroups when compared to WT animals (Fig 3F). However, only MFS+ presented a significant reduction in blood flow when compared to WT (ρ<0.0001) and MFS- (ρ<0.03) (Fig 3F). The overall findings suggest that by increasing their cardiac rates MFS- animals are still able to preserve the adequate aortic blood flow demand like WT animals. However, in MFS+ animals the increase in heart rate is not enough to provide an adequate aortic blood flow, suggesting the presence of high-output cardiac failure.

### 3.8 Heart hystology

We measured the total area of the heart, thickness of right and left ventricle walls and the area of their lumens (Fig 4). There were no differences in total heart area among all groups (WT, MFS- and MFS+). However, MFS+ animals showed a significant increase in thickness of right and left ventricle-walls when compared to WT and MFS- groups, indicating a process of concentric cardiac hypertrophy. Moreover, we found a significantly decreased ratio of left and right ventricle lumen area in MFS+ when compared to WT and MFS- (Fig 4B). These findings corroborate the blood flow data in identifying cardiac insufficiency in MFS+ animals. Interestingly, MFS- mice presented a significant increase in thickness of right ventricle wall and decreased thickness of left ventricle wall when compared to WT.

### 4. Discussion

Despite the clinical variability of MFS, cardiovascular manifestations represent the major cause of morbidity and mortality in patients [3, 10]. Animal models that mimic the clinical phenotype are instrumental to better elucidate aorta pathophysiology and to understand risk factors that contribute to aneurisms and aortic dissection in MFS. The mgΔ^loxPneo mouse model is a dominant negative model which, in heterozygosity, shows multiple MFS clinical features such as high incidence of aneurysm and dissection, thoracic spine deformities, and phenotypic variability [2, 4, 10, 11, 12]. In this model we looked for possible inducer factors in the genesis of vascular disease in adulthood.

Considering that the main role of the aorta is to conduct blood flow under pressure to meet the metabolic and nutritional demands of the body, its medial layer with elastic fibers and other fibrils was designed to provide elasticity needed for expansion and contraction resulting from heartbeats throughout life [4,5]. Here we showed that all MFS animals present alterations in the structural components that make up the wall of the thoracic aorta. The presence of severe fragmentation and disorganization of the elastic fibers, in combination with reduced and disconnected interlamellar fibers, indicate a compromised mechanism of systolic pressure damping. Finally, the preponderance of collagen type-I fibers around the elastic fibers indicate greater stiffness of the aorta and, thus, limited compliance to absorb systolic pressures [1314151617 18].

Although these structural abnormalities were present in all MFS-mice, only animals with marked thoracic spine deformities evolved with aneurysm and/or dissection at the descending thoracic aorta. Importantly, only these MFS+ animals presented aorta tortuosity and dismorphic lumen, establishing a relation between vase tortuosity and dTAAD as reported in MFS-patients [19]. Thus, we hypothesize that the extrinsic compression of the posterior aorta wall by the deformed spine may result in redirection of the blood flow to the lateral wall of the aorta, deforming its original round shape. This in turn can lead to a turbulent blood flow, and its shear force may trigger lesions in the aortic wall.

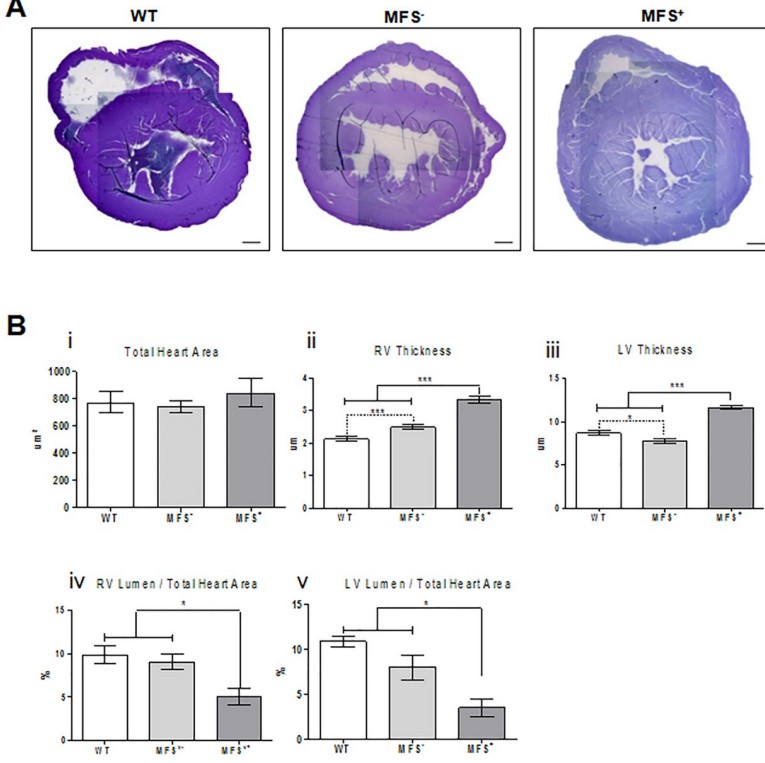

**Fig 4. Morphological analysis of the heart.** (A) Histology of a cross section of the heart of WT, MFS- and MFS+ animals. (B) Total heart area (i); thickness of (ii) the right ventricle (RV); and (iii) left ventricle (LV); proportion of right (iv) and left (v) ventricle lumen area with the total heart area of WT, MFS- and MFS+ animals. (***)$p<0.0001$; (*) $p<0.01$). Scale bars 200μm.

Corroborating with our findings, others also have pointed out the coexistence of the aorta tortuosity and aortic disease in MFS model and patients, and others correlated the high pressure and shear stress in the aorta with aberrant vortex/helix flow pattern [20, 21, 22]. Indeed, we were able to show several alterations in blood flow spectral curve in MFS-animals, in addition to increased heart rate when compared to WT-animals. However, only MFS+ mice presented decreased blood flow. Thus, the overall findings suggest that, by increasing cardiac rates, MFS- animals are able to preserve the adequate aortic blood flow demand like WT animals. However, in MFS+ animals the increase in heart rate is not enough to provide adequate aortic blood flow, suggesting the presence of high-output cardiac failure. In fact, our analysis of the heart revealed hypertrophy of both ventricles and decreased lumen-to-heart area and without difference total heart area in MFS+ animals, characteristic of cardiac insufficiency [23, 24, 25], as observed in some MFS patients [26, 27]. This important cardiac phenotype deserves further investigation.

In conclusion, we propose that vertebral thoracic compression on the wall of the aorta leads to tortuosities in the vase and turbulent blood flow which, in the presence of alterations in the matrix, triggers the formation of dTAAD of MFS mice. Thus, our results point to an important correlation between spine tortuosity and aneurysm/dissection of the descending thoracic aorta in MFS, and possibly in other diseases involving spinal deformity, which may have significant implications in the clinical management of the disease.

## Supporting information

**S1 Data Set. Data_Availability_.doc: Individual measurements used to build graphs and perform statistical analysis.**
(DOC)

## Acknowledgments

We thank the Dr. Fábio José Bonafé Sotelo and Sayonê Andrade de Moura for help with the spectral curve analysis of the aorta blood flow, and Renan Barbosa Lemes for help with the mathematical and statistics analysis.

## Author Contributions

**Conceptualization:** Rodrigo Barbosa de Souza.

**Formal analysis:** Rodrigo Barbosa de Souza, Luis Ernesto Farinha-Arcieri, Marcia Helena Braga Catroxo, Ana Maria Cristina Rebelo Pinto da Fonseca Martins, Roberto Carlos Tedesco, Luis Garcia Alonso.

**Funding acquisition:** Lygia V. Pereira.

**Supervision:** Ivan Hong Jun Koh, Lygia V. Pereira.

**Writing – original draft:** Rodrigo Barbosa de Souza, Lygia V. Pereira.

**Writing – review & editing:** Rodrigo Barbosa de Souza, Lygia V. Pereira.

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
