## [Decision Letter · Decision Letter 0]

3 Jul 2019

PONE-D-19-14919

The influence of thoracic spine deformity on aortic pathogenesis in a mouse model for Marfan syndrome

PLOS ONE

Dear Dr. Pereira,

Thank you for submitting your manuscript to PLOS ONE. After careful consideration, we feel that it has merit but does not fully meet PLOS ONE’s publication criteria as it currently stands. Therefore, we invite you to submit a revised version of the manuscript that addresses the points raised during the review process.

You need to give point-by-point responses to the reviewers' comments before the manuscript will be re-considered for publication. To enhance the reproducibility of your results, we recommend that if applicable you deposit your laboratory protocols in protocols.io, where a protocol can be assigned its own identifier (DOI) such that it can be cited independently in the future. For instructions see: http://journals.plos.org/plosone/s/submission-guidelines#loc-laboratory-protocols

We look forward to receiving your revised manuscript.

Kind regards,

Tohru Minamino, M.D., Ph.D.

Academic Editor

PLOS ONE

**Journal Requirements:**

2. Thank you for including the method of euthanasia in the Ethics Statement in your online submission form. At this time, we ask that you please revise the ethics statement in your Materials and Methods section to also include this information.

3.We note that you have stated that you will provide repository information for your data at acceptance. Should your manuscript be accepted for publication, we will hold it until you provide the relevant accession numbers or DOIs necessary to access your data. If you wish to make changes to your Data Availability statement, please describe these changes in your cover letter and we will update your Data Availability statement to reflect the information you provide.

**Comments to the Author**

1. Is the manuscript technically sound, and do the data support the conclusions?

Reviewer #1: No

Reviewer #2: No

2. Has the statistical analysis been performed appropriately and rigorously? 

Reviewer #1: Yes

Reviewer #2: Yes

3. Have the authors made all data underlying the findings in their manuscript fully available?

Reviewer #1: Yes

Reviewer #2: Yes

4. Is the manuscript presented in an intelligible fashion and written in standard English?

Reviewer #1: Yes

Reviewer #2: Yes

5. Review Comments to the Author

Reviewer #1: The influence of thoracic spine deformity on aortic pathogenesis in a mouse model for Marfan syndrome : PONE-D-19-14919

The manuscript describes further the Fbn1 mgdeltaPneo Marfan mouse model. While these mice are an interesting addition to the marfan field, I have some questions remaining.

First of all, the title does not cover the manuscript data. There are many different interesting findings which deserve the attention in this manuscript. That the spine deformity correlates with aortic disease is something the authors showed in their previous manuscript describing this model. No data is shown here to prove a causal relationship between these features. The correlation has to do with level of Fbn1 expression. However, when one improves skeletal features, the aortic features are not resolved (and vice versa) as shown by Nistala et al. (Hum Mol Genet. 2010 Dec 15;19(24):4790-8. doi: 10.1093/hmg/ddq409. Differential effects of alendronate and losartan therapy on osteopenia and aortic aneurysm in mice with severe Marfan syndrome. Nistala H et al.). Also lung problems are related to FBN1 expression and skeletal problems and aorta pathology, which may as well be of influence on aortic phenotype. So I would remove any claim of spinal deformity being the cause of a different aorta phenotype from the manuscript (it is merely a correlation) and focus on the interesting data that is actually presented, such as elastin fiber thickness, etc and the heart rate data.

It is very interesting to study the differences between the MFS v- and v+, to look for clues to use as target for further research. The cardiac data are very interesting and not well described by others I think. Could this be taken further to analyze the heart weight, histology of the hearts and study fibrosis, cardiomyocyte death, hypertrophy or dilation, Fbn1 expression in the heart or mitral valves morphology, etc.

It strikes me as strange that the aortic root of the mice have not been shown, studied, and measured, while this is the first site measured in Marfan patients. The aortic root diameter is crucial to measure in a Marfan study. Why was this not performed? The aorta pathology which is shown here is so limited compared to what may be observed in the aortic root and in other Marfan mouse models. V- and V+ aortic root data seem crucial and is lacking here.

Reviewer #2: The Manuscripts presents a microstructural analysis of the aorta in a mouse model of Marfan syndrome (MFS), focusing primarily on elastic fibers, collagen fibers, and (rather generic) interlaminar fibers. Furthermore, the Manuscript quantifies the degree of spinal deformity in the same mice, in the attempt to establish a correlation between vascular and skeletal phenotypes in MFS. Finally, the Manuscript investigates flow disturbances in the phrenic artery by means of vascular ultrasound. While some of the results are interesting, it is my opinion that the Manuscript in the present form should not be considered for publication. My major comments to the Authors are below.

METHODS: Overall, methods are rather vague and need to be explained in more detail. It is unclear how many animals were used for each different measurement.

Lines 78-79: “Thirty 6-month-old female mice […] were used. Of these, 10 animals were Wild type and 24 were heterozygous […]”. 24+10 = 34 animals overall, not 30. What was the rationale for using female mice?

Line 85: “Thoracic aorta samples (TIII-TVIII region)”. Please specify what TIII-TVIII mean. Is this the region where the MFS vascular phenotype occurs in these mice?

Line 87: “Cut at 4um”. I am assuming it means 4um thick slices were cut from the resin-embedded blocks. Please rephrase.

Lines 89-90: “at 4 equidistant points of the aorta”. Please specify where.

Line 102: “Mice were fixed in lateral decubitus”. How did the fixation process occur? On how many samples? Was the procedure performed on the same mice from which the aorta was excised?

Line 104: “Kyphosis Index Ratio”. Though references are provided, please briefly explain the meaning of this metrics and how to measure it.

Line 109: “The abdominal aorta was dissected above the infra-phrenic artery”. What do the Authors mean by dissected? MFS affect the thoracic aorta first, where the elastin fibers are more abundant. Why did the Authors decide to measure flow in the abdominal aorta and what is the relevance to the vascular phenotype in MFS?

Line 121: “Histological analysis of the thoracic aorta detected presence of vascular disease”. Please define how “vascular disease” was identified and what criteria were used to compare MFS vs. control vessels.

Line 122-123: The Authors distinguish between “aneurysmal” and “dissected” arteries. What was this classification based on? Is there a false lumen in the dissected samples?

Figure 1: The IEL seems to be fragmented in both vessels, not only the one classified as “aortic dissection”. Again, more details on the classification criteria are needed.

Line 135: Please explain the meaning of “tri-laminations”.

Line 175: What is the nature of the inter-lamellar fibers? How did the Authors define and quantify “adherence”?

Figure 3A: Though the authors did not describe the method to measure the KI, it seems to be based on the angle between the tilted vertebrae. Angle-based metrics are highly dependent on position. The animals in the three radiographs have front and hind limbs oriented differently, which may affect the angle of the spine. The Authors should ensure consistency in the positioning of the animals before measuring the KI.

Lines 186-187: “These findings establish a correlation between vascular disease and thoracic vertebral deformity”. The data presented by the Authors at best suggest a connection between vascular and skeletal disorders in MFS. To claim such a correlation, the Authors should describe mathematically the relationship between vascular and skeletal defects, though it would only confirm existing clinical observations. Indeed, from a clinical perspective, it is well known that the severity of MFS manifestations is comparable in the musculoskeletal, ocular, and vascular systems, i.e., patients with a severely deformed spine are likely to have severe vascular and ocular phenotypes as well.

Line 196: “MFSv+ presented dysmorphic aortas”. Were the aortas fixed unloaded or pressurized? It is extremely common for dilated vessels to collapse into an oval cross section when fixed unloaded. Regardless, it is the shape of the vessel in vivo that could affect blood flow. How did the cross-section look like in vivo?

Lines 211-213: “Visual inspection of the anatomy of the thoracic aorta showed that the MFSv+ subgroup presented greater aorta tortuosity”. How did the Authors define and quantified tortuosity?

Figure 3D: Please assign WT, MFSv-, MFSv+ labels to the three graphs.

Lines 249-251: “its medial layer with elastic fibers and other fibrils was designed to provide the structural barrier to expansion and contraction resulting from heartbeats through life”. The elastic fibers in the media are responsible for the elastic (no energy loss) deformation of the wall in systole and the recoil in diastole, rather than function as a structural barrier to prevent the expansion. The collagen fibers in the adventitia layer provide that barrier.

Lines 255-256: “the reduced and disconnected interlamellar fibers contribute to the unstable stage of the aortic mid-layer barrier”. It is unclear what evidence supports the claim that the aortic mid-layer barrier is unstable, or what the mid-layer barrier is to begin with.

Line 257: “greater rigidity of the aorta”. A rigid body in mechanics is a body that does not deform. Please change rigidity to “stiffness”.

Line 264-266: “the extrinsic compression of the posterior aorta by the deformed spine may result in redirection of the blood flow to the lateral wall of the aorta, shifting its original round shape to dilatation”. Again, no evidence is presented that the eccentricity of the aorta increases in vivo. Also, in MFS the ascending aorta is usually the first vessel to dilate, close to the aortic root. The ascending aorta is not in contact with the spine.

Line 266-267: “dilatation, tortuosity, and deformity can change the laminar flow to the swirling flow and trigger lesions resulting from the increase in shear force”. Swirling flow is not a scientific way to describe turbulent flow or flow vortices. Which of these two very different concepts are the Authors referring to?

Line 275: “Only MFSv+ mice presented decreased blood flow”. The Authors measure blood flow in the abdominal aorta but microstructure and tortuosity in the thoracic aorta. How can findings on these two regions be reconciled?

Line 280-282: “In conclusion, we propose that vertebral thoracic compression on the wall of the aorta can gradually modify the shape of the vase over time and result in a turbulent blood flow capable of initiating vascular pathogenesis in MFS”. The conclusion is purely speculative and not supported by experimental data. The Authors imply that skeletal deformity occurs before and is responsible for vascular disease in MFS, while clinical observations seem to suggest that the skeletal and vascular phenotypes in MFS progress at the same pace. Also, this does not apply to the ascending thoracic aorta, which is not supported by the spine, but yet is the region where aneurysms form first in MFS.

References: Given the abundance of literature on MFS, 13 references do not even begin to cover previous work.

Overall, there are many typos and grammar mistakes that must be corrected.

6. PLOS authors have the option to publish the peer review history of their article (what does this mean?). If published, this will include your full peer review and any attached files.

Reviewer #1: No

Reviewer #2: No

---

## [Author Response · Author response to Decision Letter 0]

29 Jul 2019

Reviewer #1: The influence of thoracic spine deformity on aortic pathogenesis in a mouse model for Marfansyndrome : PONE-D-19-14919

The manuscript describes further the Fbn1 mgdeltaPneoMarfan mouse model. While these mice are an interesting addition to the marfan field, I have some questions remaining.

First of all, the title does not cover the manuscript data. There are many different interesting findings which deserve the attention in this manuscript. 

Thank you for the suggestion, we have changed the title accordingly.

That the spine deformity correlates with aortic disease is something the authors showed in their previous manuscript describing this model. No data is shown here to prove a causal relationship between these features. The correlation has to do with level of Fbn1 expression. However, when one improves skeletal features, the aortic features are not resolved (and vice versa) as shown by Nistala et al. (Hum Mol Genet. 2010 Dec 15;19(24):4790-8. doi: 10.1093/hmg/ddq409. Differential effects of alendronate and losartan therapy on osteopenia and aortic aneurysm in mice with severe Marfan syndrome.Nistala H et al.). Also lung problems are related to FBN1 expression and skeletal problems and aorta pathology, which may as well be of influence on aortic phenotype. So I would remove any claim of spinal deformity being the cause of a different aorta phenotype from the manuscript (it is merely a correlation) and focus on the interesting data that is actually presented, such as elastin fiber thickness, etc and the heart rate data.

 We agree that no causal effect is shown, and we meant to say we found a correlation between the two phenotypes. We made several changes in the text in order to make that point more clear. 

 Indeed, in our original article (Fernandes et al., 2016) we observed a correlation between the severity of phenotypes among heterozygotes at 3 months of age, where the only cardiovascular phenotype analyzed was thickness of the tunica media of the aorta. In the present work, we addressed the issue of why at 6 months of age some animals develop aneurysm/dissection of the aorta while others do not. For that, we analyzed several phenotypes and identified spine tortuosity as the only one that differentiates the two groups. We modified the text to make this point more clear.

Fernandes GR, Massironi SM, Pereira LV. Identification of Loci Modulating the Cardiovascular and Skeletal Phenotypes of Marfan Syndrome in Mice. Sci Rep.2016;6:22426. doi: 10.1038/srep22426.

It is very interesting to study the differences between the MFS v- and v+, to look for clues to use as target for further research. The cardiac data are very interesting and not well described by others I think. Could this be taken further to analyze the heart weight, histology of the hearts and study fibrosis, cardiomyocyte death, hypertrophy or dilation, Fbn1 expression in the heart or mitral valves morphology, etc.

It strikes me as strange that the aortic root of the mice have not been shown, studied, and measured, while this is the first site measured in Marfan patients. The aortic root diameter is crucial to measure in a Marfan study. Why was this not performed? The aorta pathology which is shown here is so limited compared to what may be observed in the aortic root and in other Marfan mouse models. V- and V+ aortic root data seem crucial and is lacking here.

Thank you for the suggestion. We have added additional data on the heart of the animals (Figure 4). Although animals from the 3 groups (WT, MFSv- and MFSv+) did not show a difference in the total heart area, MFSv+ animals showed a significant increase of the muscular wall in both ventricles and a significant reduction in the lumen of left and right ventricles when compared to WT and MFSv-. Thus, we suggest the presence of concentric hypertrophy in MFSv+ animals, which can lead to heart failure. 

As for analyzing the aortic root, we agree that it is a critical point for MFS, and will perform a detailed study of that structure in the near future. In the meantime, we believe our findings in the thoracic aorta and the relationship between skeletal phenotype and aortic aneurysm/dissection in this mouse model is a significant contribution to the field.

Reviewer #2: 

The Manuscripts presents a microstructural analysis of the aorta in a mouse model of Marfan syndrome (MFS), focusing primarily on elastic fibers, collagen fibers, and (rather generic) interlaminar fibers. Furthermore, the Manuscript quantifies the degree of spinal deformity in the same mice, in the attempt to establish a correlation between vascular and skeletal phenotypes in MFS. Finally, the Manuscript investigates flow disturbances in the phrenic artery by means of vascular ultrasound. While some of the results are interesting, it is my opinion that the Manuscript in the present form should not be considered for publication. My major comments to the Authors are below.

METHODS: Overall, methods are rather vague and need to be explained in more detail. It is unclear how many animals were used for each different measurement.

Lines 78-79: “Thirty 6-month-old female mice […] were used. Of these, 10 animals were Wild type and 24 were heterozygous […]”. 24+10 = 34 animals overall, not 30. What was the rationale for using female mice?

We apologize for the mistake; we have corrected the text accordingly and added more details to our material and methods.

We used female mice because, although we have not formally documented this, we observe that males die earlier than females, indicating differences in disease severity between sexes. In the Fbn1C1039G MFS-model, males have been shown to have more severe aortic phenotype than females (Jiménez-Altayó et al., 2017). The same has been observed in humans (Renard et al., 2017).

Jiménez-Altayó F, Siegert AM, Bonorino F, Meirelles T, Barberà L, Dantas AP, Vila E, Egea G. Differences in the Thoracic Aorta by Region and Sex in a Murine Model of Marfan Syndrome. Front Physiol. 2017 Nov 15;8:933. doi:10.3389/fphys.2017.00933.

Renard M, Muiño-Mosquera L, Manalo EC, Tufa S, Carlson EJ, Keene DR, et al. (2017) Sex, pregnancy and aortic disease in Marfan syndrome. PLoS ONE 12(7): e0181166. https://doi.

org/10.1371/journal.pone.0181166

Line 85: “Thoracic aorta samples (TIII-TVIII region)”. Please specify what TIII-TVIII mean. Is this the region where the MFS vascular phenotype occurs in these mice?

The symbol TIII-TVIII represents the region between the third and the eighth thoracic (T) vertebrae. We have elected this portion because the major spine deformity is located in this region in all MFS animals. We changed the text to make this clearer.

Line 87: “Cut at 4um”. I am assuming it means 4um thick slices were cut from the resin-embedded blocks. Please rephrase.

We apologize for the wording; we have corrected the text accordingly.

Lines 89-90: “at 4 equidistant points of the aorta”. Please specify where.

We changed the text in order to make it clearer that for each cross-section of the aorta, we captured images of four distinct points to make measurements. 

Line 102: “Mice were fixed in lateral decubitus”. How did the fixation process occur? On how many samples? Was the procedure performed on the same mice from which the aorta was excised?

Line 104: “Kyphosis Index Ratio”. Though references are provided, please briefly explain the meaning of this metrics and how to measure it.

 We rewrote that session of material and methods to make these points clear. Briefly, the same animals were used for the different phenotyping procedures. Before sacrifice, all animals were anesthetized and position in a lateral decubitus with the aid of adhesive tape. The limbs (hind and forelimbs) were placed in moderate extension. In all process, we have taken care to avoid overextension or flexion of limbs. This could be confirmed when radiographs were analyzed because the femurs and humeri were close to parallel and perpendicular to the long axis of the spine. The radiographs that did not meet these criteria were excluded from the analysis. 

We describe the calculation of the KI, and we improved the quality of figure 3A, where we show the lines traced in the procedure.

Line 109: “The abdominal aorta was dissected above the infra-phrenic artery”. What do the Authors mean by dissected? MFS affect the thoracic aorta first, where the elastin fibers are more abundant. Why did the Authors decide to measure flow in the abdominal aorta and what is the relevance to the vascular phenotype in MFS?

The term "dissection" refers to a careful surgical step to isolate the portion of the aorta attached to the spine ligaments without lesion of the aorta wall, which was executed by microsurgical procedures under surgical microscope. This was necessary to place the ultrasound probe around the vessel. We modified the text to make this point clearer.

We agree that the thoracic aorta is first affected in MFS. However, we measured the blood flow in the abdominal aorta because, due to the tortuosity of the thoracic vessel, there was not enough space to place the ultrasound probe around it. Nevertheless, since we placed the probe in a region of the abdominal aorta close to the aorta hiatus, before the first ramification of the aorta abdominal (the infra-phrenic artery), blood flow in this region reflects blood flow of the thoracic region. We added this information to the text.

Line 121: “Histological analysis of the thoracic aorta detected presence of vascular disease”. Please define how “vascular disease” was identified and what criteria were used to compare MFS vs. control vessels.

We defined “vascular disease” as the presence of aneurysm (fusiform or saccular form) and/or aortic dissection, and these alterations were not found in samples from wild type animals – we included this explanation in the text. 

Line 122-123: The Authors distinguish between “aneurysmal” and “dissected” arteries. What was this classification based on? Is there a false lumen in the dissected samples?

Saccular aneurysms are defined as a degenerative process in the aorta wall in bleb form (Kummar et al., 2010). Aorta dissection was defined by De Bakey et al. (1961) as the rupture of the tunica intima in combination with hemorrhagic dissection in tunica media, dividing the vessel into two columns: vessel and false lumen. In the study, the dissected aorta showed false lumen and rupture of the tunica intima. We modified the text to make these points more clear. 

Kumar V, Abbas A, Aster J.Robbins Basic Pathology. 10th ed. Philadelphia: Elsevier; 2017.

De Bakey ME; Henley WS; Cooley DA, Crawford ES; Morris GC.Surgical management of dissecting aneurysms of aorta.American Heart Association.1961; 24: 290-303.

Figure 1: The IEL seems to be fragmented in both vessels, not only the one classified as “aortic dissection”. Again, more details on the classification criteria are needed.

Yes, both aneurysmal and dissecting sections of the aorta of MFS-mice present fragmented IEL. However, per definition, only dissecting aorta presents rupture of the intima lamella. 

Line 135: Please explain the meaning of “tri-laminations”.

Tri-lamination is a histology term to describe a type of fragmentation that leads to a fiber with several thinner lamella. We excluded the term for better clarity.

Line 175: What is the nature of the inter-lamellar fibers? How did the Authors define and quantify “adherence”?

 The inter-lamellar fiber is a fine aldehyde-rich fiber attached to the elastic fibers, described by Rosenquist and McCoy (1987). They are visualized by PAS histochemical technique. To quantify adherence of inter-lamellar to elastic fibers, we calculated the proportion of the inter-lamellar fibers attached and not attached to the curvature of the elastic fiber in each aorta micro-photograph (Figure below). We can add this figure as supplemental material if you find it necessary.

Figure. Identification of inter-lamellar fiber attachment. Inter-lamellar fibers attached (black arrow) and not attached (blue arrow) to the curvature of the elastic fibers are shown. Scale bar 10μm. 

Figure 3A: Though the authors did not describe the method to measure the KI, it seems to be based on the angle between the tilted vertebrae. Angle-based metrics are highly dependent on position. The animals in the three radiographs have front and hind limbs oriented differently, which may affect the angle of the spine. The Authors should ensure consistency in the positioning of the animals before measuring the KI.

Thank you pointing this out. We have explained above the methodology, and how we control positioning of the animals.

Lines 186-187: “These findings establish a correlation between vascular disease and thoracic vertebral deformity”. The data presented by the Authors at best suggest a connection between vascular and skeletal disorders in MFS. To claim such a correlation, the Authors should describe mathematically the relationship between vascular and skeletal defects, though it would only confirm existing clinical observations. Indeed, from a clinical perspective, it is well known that the severity of MFS manifestations is comparable in the musculoskeletal, ocular, and vascular systems, i.e., patients with a severely deformed spine are likely to have severe vascular and ocular phenotypes as well.

Thank you for the suggestion. For the mathematical description, we applied the point biserial correlation, which is a special Pearson's correlation coefficient that combines categorical (presence or absence vascular disease) and continuous (KI) variables. We found a correlation of -0,91 (p<0,0001), which indicates that the more severe the defect of the spine, the greater the chance of vascular disease in the thoracic aorta. We added these results to the text.

Note that although there is a clinical observation of comparable severity of the phenotypes in MFS patients, what we are describing here is different: animals with the same matrix alterations in the aortic wall have distinct outcomes when it comes to aneurysm/dissection. And the only difference we could find between the two groups is the presence of spine tortuosity in the group with aneurysm/dissection. Thus, we propose that spine tortuosity may be an important coadjuvant in the development of aneurysm/dissection.

Line 196: “MFSv+ presented dysmorphic aortas”. Were the aortas fixed unloaded or pressurized? It is extremely common for dilated vessels to collapse into an oval cross section when fixed unloaded. Regardless, it is the shape of the vessel in vivo that could affect blood flow. How did the cross-section look like in vivo?

Thank you for razing this point. Aortas were fixed unloaded, and indeed this can lead to deformities in its original shape. However, note that only aortas from animals with aneurysm/dissection showed the dysmorphism, despite being processed in the same way as those from WT and MFSv- animals. 

Nevertheless, we perfused three MFS-animals from each group and analyzed histological sections containing aorta and thoracic spine (Figure below). In that smaller sample we show the dismorphic shape of the aorta in the MFSv+ animals. We added these data to the revised manuscript.

Lines 211-213: “Visual inspection of the anatomy of the thoracic aorta showed that the MFSv+ subgroup presented greater aorta tortuosity”. How did the Authors define and quantified tortuosity?

 We scored presence/absence of tortuosity of the aortas by visual inspection and comparison among animals. Although somewhat subjective, we could identify two groups with and without severe curves in the aorta, as illustrated in Figure 3C.We did not quantify aorta tortuosity.

 The presence of aortic tortuosity in animals with spine tortuosity may not be surprising, since the thoracic aorta descends continuously and very close to the thoracic spine. We modified the text accordingly.

Figure 3D: Please assign WT, MFSv-, MFSv+ labels to the three graphs.

We apologize for the missing labels, they have been assigned to the corrected Figure 3.

Lines 249-251: “its medial layer with elastic fibers and other fibrils was designed to provide the structural barrier to expansion and contraction resulting from heartbeats through life”. The elastic fibers in the media are responsible for the elastic (no energy loss) deformation of the wall in systole and the recoil in diastole, rather than function as a structural barrier to prevent the expansion. The collagen fibers in the adventitia layer provide that barrier.

Thank you for pointing this out, we have changed the text accordingly. 

Lines 255-256: “the reduced and disconnected interlamellar fibers contribute to the unstable stage of the aortic mid-layer barrier”. It is unclear what evidence supports the claim that the aortic mid-layer barrier is unstable, or what the mid-layer barrier is to begin with.

Line 257: “greater rigidity of the aorta”. A rigid body in mechanics is a body that does not deform. Please change rigidity to “stiffness”.

Thank you for the comments and corrections, we have changed the text accordingly.

Line 264-266: “the extrinsic compression of the posterior aorta by the deformed spine may result in redirection of the blood flow to the lateral wall of the aorta, shifting its original round shape to dilatation”. Again, no evidence is presented that the eccentricity of the aorta increases in vivo. Also, in MFS the ascending aorta is usually the first vessel to dilate, close to the aortic root. The ascending aorta is not in contact with the spine.

Thank you for the comments. We included data showing that dysmorphism of the aorta in vivo in MFSv+ animals. Indeed, the ascending aorta receives the total force of the circulation dynamics, and probably for that reason, it is the first part of the aorta to start a dilatation process in MFS. We will perform a detailed study of that structure, as well as the aortic root in the near future. In the meantime, we believe our findings in the thoracic aorta and the relationship between skeletal phenotype and aortic aneurysm/dissection in this mouse model is a significant contribution to the field.

Line 266-267: “dilatation, tortuosity, and deformity can change the laminar flow to the swirling flow and trigger lesions resulting from the increase in shear force”. Swirling flow is not a scientific way to describe turbulent flow or flow vortices. Which of these two very different concepts are the Authors referring to?

We apologize for the inadequate wording. We referred to turbulent and we corrected the text. 

Line 275: “Only MFSv+ mice presented decreased blood flow”. The Authors measure blood flow in the abdominal aorta but microstructure and tortuosity in the thoracic aorta. How can findings on these two regions be reconciled?

As explained above, due to technical limitations we were not able to measure blood flow in the thoracic aorta. However, the measurements performed at the abdominal aorta, before the first ramification (inferior phrenic artery), reflect the dynamic of the thoracic aorta. Thus, we used the analysis of blood flow in the abdominal aorta as a proxy for the thoracic aorta.

Line 280-282: “In conclusion, we propose that vertebral thoracic compression on the wall of the aorta can gradually modify the shape of the vase over time and result in a turbulent blood flow capable of initiating vascular pathogenesis in MFS”. The conclusion is purely speculative and not supported by experimental data. The Authors imply that skeletal deformity occurs before and is responsible for vascular disease in MFS, while clinical observations seem to suggest that the skeletal and vascular phenotypes in MFS progress at the same pace. Also, this does not apply to the ascending thoracic aorta, which is not supported by the spine, but yet is the region where aneurysms form first in MFS.

We have rephrased our conclusions in order to be clearer. Regardless of the timing of development of spine and aorta deformities, we believe that our data supports a model in which deformities of the vase, possibly caused by deformities of the spine, lead to turbulent blood flow which in turn, in the presence of alterations in the matrix of the aortic wall, leads to aortic aneurysm/dissection in MFS-mice. 

References: Given the abundance of literature on MFS, 13 references do not even begin to cover previous work.

We agree, and we have included the most relevant and recent references on the different topics. 

Overall, there are many typos and grammar mistakes that must be corrected.

 We apologize for the typos and mistakes. The text has been thoroughly reviewed.

---

## [Decision Letter · Decision Letter 1]

16 Sep 2019

PONE-D-19-14919R1

Association of thoracic spine deformity and cardiovascular disease in a mouse model for Marfan syndrome

PLOS ONE

Dear Dr. Pereira,

Thank you for submitting your manuscript to PLOS ONE. After careful consideration, we feel that it has merit but does not fully meet PLOS ONE’s publication criteria as it currently stands. Therefore, we invite you to submit a revised version of the manuscript that addresses the points raised during the review process.

We would appreciate receiving your revised manuscript by Oct 31 2019 11:59PM. To enhance the reproducibility of your results, we recommend that if applicable you deposit your laboratory protocols in protocols.io, where a protocol can be assigned its own identifier (DOI) such that it can be cited independently in the future. For instructions see: http://journals.plos.org/plosone/s/submission-guidelines#loc-laboratory-protocols

We look forward to receiving your revised manuscript.

Kind regards,

Tohru Minamino, M.D., Ph.D.

Academic Editor

PLOS ONE

Additional Editor Comments (if provided):

Because the revised paper somewhat remains preliminary, we asked an additional reviewer to evaluate the manuscript. Several issues have been raised by Reviewer #3, and we believed those should be addressed for publication. Thus, the Editor decided to require a major revision.

Reviewers' comments:

Reviewer's Responses to Questions

**Comments to the Author**

1. If the authors have adequately addressed your comments raised in a previous round of review and you feel that this manuscript is now acceptable for publication, you may indicate that here to bypass the “Comments to the Author” section, enter your conflict of interest statement in the “Confidential to Editor” section, and submit your "Accept" recommendation.

Reviewer #1: All comments have been addressed

Reviewer #2: All comments have been addressed

Reviewer #3: (No Response)

2. Is the manuscript technically sound, and do the data support the conclusions?

Reviewer #1: Yes

Reviewer #2: (No Response)

Reviewer #3: Partly

3. Has the statistical analysis been performed appropriately and rigorously? 

Reviewer #1: Yes

Reviewer #2: I Don't Know

Reviewer #3: Yes

4. Have the authors made all data underlying the findings in their manuscript fully available?

Reviewer #1: Yes

Reviewer #2: Yes

Reviewer #3: Yes

5. Is the manuscript presented in an intelligible fashion and written in standard English?

Reviewer #1: Yes

Reviewer #2: Yes

Reviewer #3: Yes

6. Review Comments to the Author

Reviewer #3: This manuscript presents the association of the thoracic kyphosis and the incidence of descending aortic aneurysm/dissection, using their kyphosis-prone Marfan mice (mgΔloxPneo: in-frame deletion of FBN1 exon 19-24). As authors mentioned, risk factors for type B aortic dissection remain elusive (line 297), and thus this study might provide practical information for prognostic stratification of patients with Marfan syndrome (MFS), however, data presentation is insufficient for drawing scientific conclusions. I have serious concerns on several issues in this paper.

(1) I was confused with the word of “vascular disease (defined at line 142)” used for “the presence of thoracic aneurysm and/or dissection”. It is better to use the word of “descending thoracic aortic aneurysm and/or dissection (dTAAD)” or something, because the mgΔloxPneo mice seem to have significant histological changes in entire aorta.

(2) Authors should show the objective definition and data for the “presence of descending aortic aneurysm (line 142)”. It might be difficult to evaluate due to the “dysmorphism of the aorta” in kyphosis mice, however, authors must show/compare the aortic lumen area when dividing into two groups. Fig. 1a, c are inappropriate images showing the presence of aneurysm.

(3) Authors show the positive correlation between the presence of dTAAD (categorical) and kyphosis index (KI) (continuous), however, I think authors should show the correlation between the lumen area and KI, as I described above. Hopefully, it would be preferable to present the time serial data (e.g. at 3 months), because the mgΔloxPneo mice possibly develop kyphosis at more earlier ages than 6 months.

(4) Ref 19 (Int J Cardiol 2015;194:7-12) shows the relation between vase tortuosity and aortic disease in patients with MFS (line 314), however, “aortic root dilatation” is defined as aortic disease in Ref 19. Authors should rather show the data of aortic root diameter as a control as requested by reviewer #1. It is important to examine the influence of the kyphosis on the location of aortic aneurysm and dissection.

(5) Authors show the hemodynamic data obtained by aortic blood blow and cardiac histological analyses. However, it seems inaccurate for drawing any conclusions in the present methods; heart rates were very slow and cardiac index should be obtained by echocardiography. Especially, it is very important to analyze the cardiac phenotypes of mgΔloxPneo mice by echocardiography and/or transcriptome analysis. Previously reported clinical and experimental Marfan hearts develop dilated cardiomyopathy (DCM)-like phenotypes after cardiac loading, but not concentric hypertrophy in this study. It’s possible, but should be evaluated by accurate modalities in the future. I do not think, hemodynamic data is necessarily included in this study, mainly showing the association of the thoracic kyphosis and the incidence of descending aortic aneurysm/dissection.

Minor points;

(6) It is better to explain more mgΔloxPneo mice. (e.g. in-frame deletion of FBN1 exon 19-24; susceptibility of the kyphosis).

(7) Line 143, 151; aneurism

(8) Comma in the numbers in the part of method.

7. PLOS authors have the option to publish the peer review history of their article (what does this mean?). If published, this will include your full peer review and any attached files.

Reviewer #1: No

Reviewer #2: No

Reviewer #3: No

---

## [Author Response · Author response to Decision Letter 1]

3 Oct 2019

Reviewer #3: This manuscript presents the association of the thoracic kyphosis and the incidence of descending aortic aneurysm/dissection, using their kyphosis-prone Marfan mice (mgΔloxPneo: in-frame deletion of FBN1 exon 19-24). As authors mentioned, risk factors for type B aortic dissection remain elusive (line 297), and thus this study might provide practical information for prognostic stratification of patients with Marfan syndrome (MFS), however, data presentation is insufficient for drawing scientific conclusions. I have serious concerns on several issues in this paper.

(1) I was confused with the word of “vascular disease (defined at line 142)” used for “the presence of thoracic aneurysm and/or dissection”. It is better to use the word of “descending thoracic aortic aneurysm and/or dissection (dTAAD)” or something, because the mgΔloxPneo mice seem to have significant histological changes in entire aorta.

Thank you for the suggestion, we changed the nomenclature to dTAAD.

(2) Authors should show the objective definition and data for the “presence of descending aortic aneurysm (line 142)”. It might be difficult to evaluate due to the “dysmorphism of the aorta” in kyphosis mice, however, authors must show/compare the aortic lumen area when dividing into two groups. Fig. 1a, c are inappropriate images showing the presence of aneurysm.

The presence or absence of descending aortic aneurysm was defined by the histologic aspect and confirmed by two different pathologists. In this study, we found the saccular aneurysm, which is defined as a degenerative process in the aorta wall in bleb form (Kummar et al. 2010). We thank you for the suggestion and we changed the picture for better visualization of the aneurysm. 

In addition, we included the measurement of the area of the aortic lumen in the three groups (WT, MFS+, and MFS-), in Figure 1. We observed a significant decreased of aorta's lumen in MFS+ when compared to WT and MFS- animals.

(3) Authors show the positive correlation between the presence of dTAAD (categorical) and kyphosis index (KI) (continuous), however, I think authors should show the correlation between the lumen area and KI, as I described above. Hopefully, it would be preferable to present the time serial data (e.g. at 3 months), because the mgΔloxPneo mice possibly develop kyphosis at more earlier ages than 6 months.

Thank you for the suggestion. We used Pearson's Correlation and observed a positive correlation between KI and area of the aorta's lumen (cor 0,7984, p<0,0001). Thus, the larger the spine tortuosity (lower KI), the lower the lumen area. We added this to the text and figure 3D.

Although the suggestion of the time serial analysis is very interesting, the focus of the present study was to characterize the cardiovascular phenotypes at an age where most of the animals showed some alteration, i.e., at 6 months (Lima et al., 2010). From that analysis we found the variability in dTAAD occurrence and went on to try to understand its origin. 

(4) Ref 19 (Int J Cardiol 2015;194:7-12) shows the relation between vase tortuosity and aortic disease in patients with MFS (line 314), however, “aortic root dilatation” is defined as aortic disease in Ref 19. Authors should rather show the data of aortic root diameter as a control as requested by reviewer #1. It is important to examine the influence of the kyphosis on the location of aortic aneurysm and dissection.

Indeed, in this work we focused in the descending aorta due to the frequent finding of dissection and aneurysms in descending thoracic aorta (type B) in MFS patients. Please note that in reference 19, in addition to correlating aortic tortuosity to aortic root diameter, the authors also showed that those patients with higher aortic tortuosity had a higher probability of developing aortic dissection (“After 49.3 ± 8.8 months follow-up, 33 patients met the combined clinical endpoint (7 dissections) with a significantly higher ATI at baseline than patients without endpoint (1.98 ± 0.2 vs. 1.86 ± 0.2, p=0.002). Patients with an ATI>1.95 had a 12.8 times higher probability of meeting the combined endpoint (log rank-test, p<0.001) and a 12.1 times higher probability of developing an aortic dissection (log rank-test, p=0.003) compared to patients with an ATI<1.95. “). 

They go on to suggest that quantification of aortic tortuosity may improve prediction of type B aortic dissection. We were referring to that aortic phenotype (dissection), not to the increase in aortic root diameter, and thus we have changed our text accordingly to make this point more clear. 

 (5) Authors show the hemodynamic data obtained by aortic blood blow and cardiac histological analyses. However, it seems inaccurate for drawing any conclusions in the present methods; heart rates were very slow and cardiac index should be obtained by echocardiography. Especially, it is very important to analyze the cardiac phenotypes of mgΔloxPneo mice by echocardiography and/or transcriptome analysis. Previously reported clinical and experimental Marfan hearts develop dilated cardiomyopathy (DCM)-like phenotypes after cardiac loading, but not concentric hypertrophy in this study. It’s possible, but should be evaluated by accurate modalities in the future. I do not think, hemodynamic data is necessarily included in this study, mainly showing the association of the thoracic kyphosis and the incidence of descending aortic aneurysm/dissection.

Thank you for the careful analysis of our data. We chose to include the hemodynamic data in order to support our hypothesis of disturbed blood flow in the animals with dTAAD/low KI. The analysis of the heart was a suggestion from one of the reviewers based on our suggestion of high-output cardiac failure, and revealed another important phenotype of this MFS model. 

The low values of the heart rates are probably due to the anesthetic procedure. However, although the heart rates were slow, all animals were submitted to the same procedure, thus, the differences found are significant. 

Indeed DCM-like phenotypes are present in some MFS patients and in the hypomorphic model mgR (Cook et al., 2014). Nevertheless, LV hypertrophy has also been described in MFS patients and in a different mouse model for the disease (Tae et al. 2016; Arumamate et al. 2018). We agree that echocardiography and/or trancriptome analysis of the heart will provide more insights on the cardiac phenotype, and we should perform them in a future study that will focus on the cardiac phenotypes of our MFS model. In this study we presented those data to corroborate our hypothesis of high-output cardiac failure. Nevertheless, despite not being our initial goal, our histological description revealed an important cardiac phenotype that deserves further investigation. We have modified the text in order to make these points more clear.

• Cook JR, Carta L, Bénard L, Chemaly ER, Chiu E, Rao SK, Hampton TG, Yurchenco P; GenTAC Registry Consortium, Costa KD, Hajjar RJ, Ramirez F. Abnormal muscle mechanosignaling triggers cardiomyopathy in mice with Marfan syndrome. J Clin Invest. 2014;124(3):1329-39.

• Tae HJ, Petrashevskaya N, Marshall S, Krawczyk M, Talan M. Cardiac remodeling in the mouse model of Marfan syndrome develops into two distinctive phenotypes. Am J Physiol Heart Circ Physiol. 2016; 310(2):H290-9. 

• Arunamata AA, Nguyen CT, Ceresnak SR, Dubin AM, Olson IL, Murphy DJ, Selamet Tierney, ES. Utility of serial 12-lead electrocardiograms in children with Marfan syndrome. Cardiol Young. 2018;28(8):1009-1013. 

Minor points;

(6) It is better to explain more mgΔloxPneo mice. (e.g. in-frame deletion of FBN1 exon 19-24; susceptibility of the kyphosis).

Thank you for the suggestion, we included more info about the mice.

(7) Line 143, 151; aneurism

We apologize, we have corrected the text.

 (8) Comma in the numbers in the part of method.

We apologize, we have corrected the text.

---

## [Decision Letter · Decision Letter 2]

17 Oct 2019

Association of thoracic spine deformity and cardiovascular disease in a mouse model for Marfan syndrome

PONE-D-19-14919R2

Dear Dr. Pereira,

We are pleased to inform you that your manuscript has been judged scientifically suitable for publication and will be formally accepted for publication once it complies with all outstanding technical requirements.

With kind regards,

Tohru Minamino, M.D., Ph.D.

Academic Editor

PLOS ONE

Additional Editor Comments (optional):

Reviewers' comments:

Reviewer's Responses to Questions

**Comments to the Author**

1. If the authors have adequately addressed your comments raised in a previous round of review and you feel that this manuscript is now acceptable for publication, you may indicate that here to bypass the “Comments to the Author” section, enter your conflict of interest statement in the “Confidential to Editor” section, and submit your "Accept" recommendation.

Reviewer #3: All comments have been addressed

2. Is the manuscript technically sound, and do the data support the conclusions?

Reviewer #3: Yes

3. Has the statistical analysis been performed appropriately and rigorously? 

Reviewer #3: Yes

4. Have the authors made all data underlying the findings in their manuscript fully available?

Reviewer #3: Yes

5. Is the manuscript presented in an intelligible fashion and written in standard English?

Reviewer #3: Yes

6. Review Comments to the Author

Reviewer #3: The authors responded to my all concerns.

7. PLOS authors have the option to publish the peer review history of their article (what does this mean?). If published, this will include your full peer review and any attached files.

Reviewer #3: No

---

## [Editor Report · Acceptance letter]

7 Nov 2019

PONE-D-19-14919R2 

Association of thoracic spine deformity and cardiovascular disease in a mouse model for Marfan syndrome 

Dear Dr. Pereira:

I am pleased to inform you that your manuscript has been deemed suitable for publication in PLOS ONE. Congratulations! Your manuscript is now with our production department. 

With kind regards,

on behalf of

Professor Tohru Minamino 

Academic Editor

PLOS ONE